# Concretized structural evolution supported assembly-controlled film-forming kinetics in slot-die coated organic photovoltaics

Hao Zhang [1,2], Chenyang Tian [1,2], Ziqi Zhang[1,2], Meiling Xie[1,2], Jianqi Zhang [1] ✉, Lingyun Zhu [1] & Zhixiang Wei [1,2] ✉

Bulk-heterojunction structured small-area organic solar cells are approaching 20% power conversion efficiency, but the blurred film-forming kinetics in the fabrication of large-area devices causes significant PCE loss and restrains the potential of commercialization. Such blurring came from insufficient knowledge of structural evolution during the film-forming process. Here, we concretize the evolution process with structures detailed to the submolecular level by comprehensive investigations of in-situ UV-vis spectroscopy, Atomic Force Microscope, Grazing Incident Wide Angle X-ray Scattering, and molecular dynamic simulation. With such hierarchical structural knowledge, assembly-controlled film-forming kinetics is proposed to explain the whole picture. Such assembly is determined by molecule configuration and can be tuned via external conditions. Understanding this kinetics will contribute to screening large-area device fabrication conditions, and the detailed structural knowledge could inspire the future design of novel photovoltaic materials that are intrinsically excellent in large-area device fabrications.

Solution-processed organic photovoltaics (OPV) are flexible, light-weighted, and variable in a wide range of colors, which gives numerous possibilities of application like building integrated photovoltaics (BIPV)[1–3], indoor photovoltaics[4–6], and flexible electronics[7–10]. Organic solar cells (OSC) have achieved over 20% power conversion efficiency (PCE) in the laboratory and are at their predawn for wide commercial applications[10]. However, one more step still needs to be taken to bring the initial light of the day: State-of-the-art OSCs are usually fabricated using spin-coating techniques and have a light-harvesting area of <0.1 cm$^2$, which is impractically small for most scenarios.

Brilliant minds are working hard to scale up the OSCs to a commercially available size, but the process is rather difficult. PCE loss in this process is the most critical problem that needs to be addressed. Nominating several outstanding reports so far, Cao et al. applied a sequential bilayer coating technique and achieved 11.4% PCE for 1 cm$^2$ OSC device using PM6 and IT-4F, which kept 88% PCE of its spin-coated small-area device[11]. Chen et al.[12] utilized a combined annealing strategy

to optimize the contacting surface between the charge transporting layer and active layer, achieving a 14.79% PCE at ~19 cm$^2$ size, which kept 86% PCE of its small-area counterpart. But if such loss was compared with best spin-coated small-area OSCs (over 19% PCE)[13,14], only ~76% PCE was obtained in these large-area devices. One of the major deviations between a small-area OPV device and a large-area device is their film-forming kinetics during the fabrication process[15,16]. These results appeal to a comprehensive kinetic model that could explain the morphology evolution during OPV device fabrication, guide the optimization of processing conditions, and minimize such performance degradation.

Previous reports tried to solve the puzzle from multiple directions. From a general film-forming perspective, Müller-Buschbaum et al. compared several non-fullerene acceptors (NFAs) based film-forming kinetics in chloroform and chlorobenzene and yielded the conclusion that balanced crystallinity and aggregation was the key to obtaining a high-performance OPV device[17]. On the intermolecular

[1]CAS Key Laboratory of Nanosystem and Hierarchical Fabrication, National Center for Nanoscience and Technology, 100190 Beijing, China. [2]University of Chinese Academy of Sciences, 100049 Beijing, China. ✉e-mail: zhangjq@nanoctr.cn; weizx@nanoctr.cn

interaction level, Liu et al. used a second donor polymer to facilitate acceptors growing into fibril structure, leading to outstanding device efficiency[13]. From the perspective of molecular design, Zhou et al. concluded that the size of the sidechain in the pyrrole ring of Y6 analogs is crucial in controlling the aggregation of active layer[18]. A series of comparative reports from our previous work shows two promising NFA molecules named Qx-1 and Qx-2, reaching a similar 17.9% and 18.2% PCE in fabricating small-area OSC devices[19], but 1 cm² device made from PM6:Qx-2 almost didn't work. Analysis of the in-situ UV-vis absorption spectrum indicates excessive crystallinity in Qx-2-based devices which leads to early phase separation and oversized crystal domains[20]. These reports conclude the valuable experience we had on the film-forming process in the fabrication of large-area OSCs, but leave the most critical questions unanswered: What is the actual structural evolution during the film-forming process? Without such knowledge, current models can only work empirically and help little in technological migration between different OPV device fabrications or guiding molecule design.

In this work, we aimed to concretize the structural evolution with in-situ UV-vis spectroscopy, multiscale structural analysis, and molecular dynamic simulation. Three typical acceptors we investigated, named Y6, N3, and L8-BO, have assembly features determined by competition between the same backbone packing and repulsion of different sidechains which lead to significant PCE differences in slot-die coated 1 cm² device. A comprehensive analysis of our experimental results and simulations revealed that the assembly of these molecules determines the hierarchical evolution of microstructure and morphology during the film-forming process. These assemblies, especially a star-shaped trimer micro-assembly, could match up multiple experimental observations. Based on the evidence above, a comprehensive assembly-controlled film-forming kinetics model was proposed and successfully explained further spectral evolution characteristics. Our work proves the paradigm that combining molecular dynamic simulation, in-situ UV-vis spectroscopy, and multiscale structural analysis composed of Atomic Force Microscope(AFM) and Grazing Incident Wide Angle X-ray Scattering(GIWAXS) is capable of investigating complicated multiscale structural evolution, which could be an inspiration for the construction of other kinetic models.

## Results

Polymer donor PM6 and three Y6 analogs, named Y6, N3, and L8-BO, were chosen to be the subjects of our investigation. First, we used in-situ UV-vis spectroscopy to monitor the spectral evolution of the film-forming process during the slot-die coating process and found two distinct spectral evolution features within. Later, GIWAXS and AFM results suggest that the morphology of coated PM6/Y6 analog films have significant differences on a scale from several molecules to aggregates with a few hundred nanometers. Finally, molecular dynamic simulation unveiled the key dimer and microaggregate configurations which could match up with both spectral evolution features and GIWAXS scattering features. The comprehensive discussion of the evidence leads to the proposal of our kinetic model, which will be discussed in the last sub-section of our results.

### In-situ UV-vis spectrum of slot-die film-forming process

The scheme of in-situ UV-vis spectrum incorporation with slot-die apparatus is shown in Fig. 1a, and the molecular structure of materials used in this work can be found in Fig. 1b, c. Compared with Y6, N3 extended the pivot of inner alkyl chains further away from the backbone, reducing its interference with backbone-related packing[21]. While L8-BO replaced outer linear alkyl chains with branched alkyl chains, further increasing its interference with backbone[22]. By varying the alkyl chains, we had the chance to investigate solely how side chains affect the film-forming process. Though all performed excellently in the fabrication of small-area OSCs, the three NFAs performed quite differently after being made into large-area devices using *o*-xylene. As shown in Supplementary Table 1, PCEs of 1 cm² devices made from PM6/Y6 analogs are dramatically different in numbers and dependency on the substrate temperature, as shown in the table and Supplementary Fig. 1, where N3-based devices performed poorly overall, especially under low coating temperatures. Y6-based devices showed moderate performance but strong temperature dependency, varying PCE from 3% when coated at 40 °C to 7.5% coated at 100 °C. On the contrary, L8-BO-based devices depicted uniformly well performance across various temperatures, reaching a decent 12.17% PCE without the help of any solvent additive or third components. Such differences could indicate significant differences in film structures, not to mention the kinetics that result in such divergence. When compared to small-

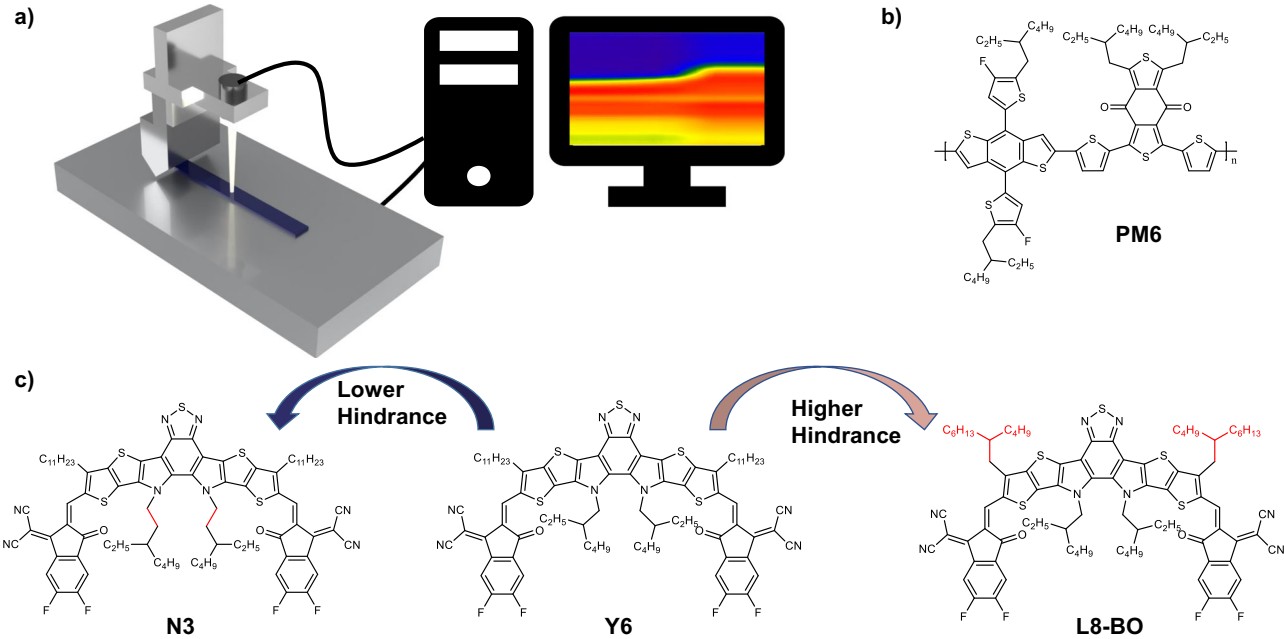

**Fig. 1 | Schematic of slot-die integrated In-situ UV-vis apparatus and material structure. a** Instrumental setup of slot-die integrated in-situ UV-vis spectrometer which works on transmission mode; structure of (**b**) polymer donor PM6, (**c**) non-fullerene acceptor N3 (left), Y6 (middle), and L8-BO (right).

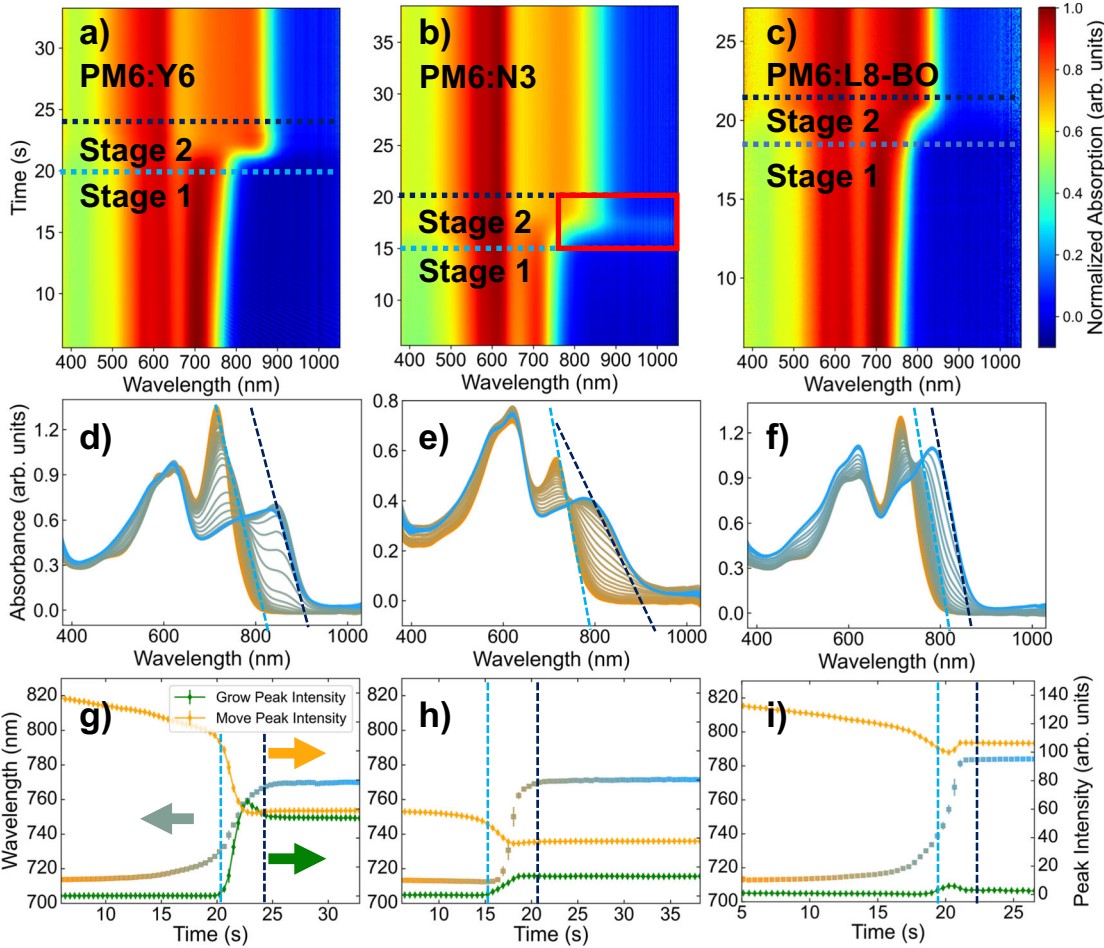

**Fig. 2 | Analysis of in-situ UV-vis spectrum of slot-die coated blends based on PM6:Y6, PM6:N3 and PM6:L8-BO.** (**a**–**c**)The waterfall plots show the spectral evolution during the film-drying process. The stacking plots (**d**–**f**) and derived fitting results (**g**–**i**) elaborate the fitting results of grow-moving two peak model. Two stages of different evolving speeds were differentiated with two dashed lines. Dashed lines in (**d**–**f**) and (**g**–**i**) mark the same segment of stages. The red box in (**b**) points out the raise of baseline in the PM6:N3 coating session. In order to clarity the figure information, the datapoints in **g**–**i** were derived from spectrum frames in the represented timespan with average being the value and standard deviation as error bar. At least 20 frames of sepctrum were represented by each point.

area devices fabricated with spin-coating technique and *o*-xylene as a solvent, as shown in Supplementary Table 2, L8-BO achieves the best performance with both small and large area devices, Y6 obtain a mediocre result, and N3 excels in a small area but failed catastrophically in large area devices.

In Fig. 2a–c we plot the in-situ UV-vis spectrum of slot-die coated blend films with substrate temperature set at 40 °C. A rapid transition in the absorption spectrum was observed in all three spectrums, but the details are different. As the solvent evaporates, Y6-based blend film in Fig. 2a exhibits a gradually red-shifted absorption edge with a sharp edge at around 21 s. The peak afterward slightly blue shifted and finally became stable. In the N3 blend film (Fig. 2b), the progressively red-shifted trend was seldom seen until the continuous edge shift between 15 s and 18 s. At around 18 s, the spectrum baseline observed an obvious rise and then decline (marked with a red box in Fig. 2b and can be distinguished in Fig. 2e), which is unique in three blends. And the lower slope of the absorption edge, i.e., the purple line in Fig. 2e, indicates a large variance in electron transition energy. The phenomenon suggests that PM6:N3 film may possess a more-closely-but-more-disordered stacking of acceptor molecules, further depicting its uniqueness compared with the other two Y6 analogs. While in L8-BO's case, a progressively red-shifted trend without observable baseline movement was observed. We repeated coatings under substrate temperature ranging from room temperature (RT) to 100 °C and plot

results in Supplementary Fig. 2. A similar evolution of spectrums is observed while the baseline shift of N3 based blend became sharper as the temperature rose. We attributed the baseline shift observed in the N3 blend to the Mie scattering of nanocrystals with a few microns diameter during their nucleation and growth[23]. Faster solvent evaporation leads to a more unified crystal grain size versus time, and results in a sharper scattering feature observed at higher temperatures[20]. When grains further grow away from a few microns, the Mie scattering effects diminish; thus, the baseline declined again. As depicted in Supplementary Fig. 3, the optical microscope of PM6:N3 blend film exhibited clear temperature-dependent crystallization, which could further be the evidence of our discussion above. Though the intensity of Mie scattering can be used to estimate the average size of scattering particles[24], such analysis is rather hard considering the film thickness and refractive index changes during the film-forming process[25]. The in-situ spectroscopy sheds light on the poor performance of N3-based devices: over-crystalizing in low temperatures results in a changed donor/acceptor ratio in films and creates numerous defects, making N3-based devices coated at a low temperature barely usable. As the temperature goes up to 100 °C, fast solvent evaporation remedied the setback and created several usable devices at still-low performance. Over crystallization and change in the ratio of donor and acceptor also reflects on the initial state of the absorption spectrum. Compared with PM6:Y6 and PM6:L8-BO, the

acceptor absorbance peak in the PM6:N3 blend is obviously lower. Such phenomenon can also be observed in spectrums coated under other temperatures.

We divide the evolution of the absorption edge into two stages with two dashed lines, as depicted in Fig. 2a–c. The difference in the evolution of the absorption edge is more obvious when inspecting the stacking plot of absorbance in Fig. 2d–f. Starting with the absorbance curve of solution paint in the orange line, the color gradually changes into blue, representing the absorbance curve of the film. We transferred the dashed line in Fig. 2a–c to Fig. 2d–f, wrapping the spectral evolution of stage two in between.

The difference is evident if we observe how the patch of lines fills up the square between the two lines: For the PM6:Y6 blend depicted in Fig. 2d, a peak rise from the bottom of the square and gradually grows up to the top. For PM6:L8-BO depicted in Fig. 2f, the absorption edge gradually shifts from the short wavelength side toward the long wavelength side, and the edge line is almost parallel in the whole process. For the PM6:N3 blend, apart from the obvious baseline shift, the line patch depicted a mixed behavior with both a growing feature and a moving feature. Moreover, in stage one, N3 might have taken the H-aggregation feature while Y6 and L8-BO take J-aggregation, which can be differentiated by the different evolution of major absorption peak in stage one, as depicted in Supplementary Fig. 4. Such a result is in accordance with previous reports on similar Y6 derivatives[26]. When applying a similar analysis to pristine acceptor films, the peak positions of all three pure films shifted further in longer wavelengths as depicted in Supplementary Fig. 5. Y6 & N3 film exhibited a clearly growing feature in stage two, while L8-BO still takes moving as the majority. However, the loose patch of L8-BO could indicate a larger portion of intensity contributed by the growing feature of L8-BO.

Summarizing the findings above, we observed two distinguished features, i.e., moving feature and growing feature, in the evolution of acceptor major absorption peak. The moving feature could indicate a continuous assembly of molecules, extending their conjugated length and lowering the band gap between Highest Occupied Molecular Orbital (HOMO) and Lowest Unoccupied Molecular Orbital (LUMO), resulting in a consecutive change in optical bandgap[27]. Although the growing feature could indicate an assembly with constant absorption characteristics was gradually generated during the film-forming process, increasing its quantity and raising the intensity of its absorption peak. With the presumption, we came up with a Grow-Move Two Peak Model to fit the evolution of acceptor absorbance edge of major absorption peak during the film-forming process. Details of the fitting can be found in Supplementary Note 1. Intriguingly we did find a growing peak, most probably located at 847 nm, together with a moving peak starting from ~715 nm could explain the evolutions acceptor absorbance edge in all blends. We fitted the spectral evolution in Fig. 2d–f and drew the peak intensity of both peaks and the position of the moving peak in Fig. 2g–i. Three snapshots captured from stage two are shown in Supplementary Fig. 6, showing the split result of the moving peak and growing peak in the beginning, halfway, and almost dried film, and confirming the validity of our fitting model.

In the Y6 blend, the growing peak appears in stage two and is rivaled by the moving peak at the end of the film-forming process, with nearly equal intensity when compared with each other. While in L8-BO's case, the moving peak dominated the whole film-forming process, with only a small lump seen for the growing peak in stage two. In N3's case, a mixed but moving lead mechanism is resolved. Such results are in accordance with our perception of spectrums. Moreover, the shifting of the absorption edge starts in stage one for the Y6 blend and the L8-BO blend, while N3's absorption edge shifts after the increment of growing peak intensity. In Y6 and N3 blend, both moving peak stops at around 764 nm. While in L8-BO's case, the moving peak continues to move beyond 780 nm. The model was later successfully fitted for the evolution of pristine film of acceptors, as we plot the fitting results in Supplementary Fig. 7.

Under the scheme of the Grow-Move Two Peak Model we found both similarities and disagreements in three Y6 analogs. Among these, Y6 is the most special one since it possesses similarities with both molecules. Y6 and N3 share the same parking position of the moving peak, which might indicate a similar conjugated length of continuous micro-assembly in films. While Y6 and L8-BO both start their shift of moving peak in phase one, which might suggest similar assembly behavior in the solution. Moreover, though N3's fitting result indicates a moving-dominate feature, the truth might be over dominating growing kinetic, which forms microcrystals that cause Mie scattering phenomenon as we have already discussed. If we put L8-BO as the representation of moving kinetic, this makes Y6 the one molecule that made a balance between the two extremes, which partially explains its similarity with both ends. Considering the moderate repulsion of Y6's sidechains, such balance is anticipated but its structural and kinetic detail are still unknown.

## Multiscale structural analysis

We performed AFM scanning on film coated by blends, and the results are shown in Fig. 3b–d. As a reference, AFM of pure PM6 film coated under the same conditions was put in Fig. 3a as well. The morphology of pure PM6 film and three blend films was in great disagreement at first glance. Pure PM6 film exhibited a clear fibril feature, while in the Y6 blend's film, a spherical structure is observed with greater fluctuation in height. N3's blend showed a similar fibril structure compared with that of PM6 film while the L8-BO blend exhibited a flake-like structure. Considering the over-crystallization of N3's blend and the current AFM image of the N3 blend was taken in the region with no grains, the local composition of donor and acceptor may vary, and the donor would be the majority component in this region. This could explain the similarity of AFM images between pure PM6 film and N3 blend. But more similarity can be found if one takes a closer look at the AFM images. On the bottom left of Fig. 2b, we plot an enlarged section of the image, showing the yarn-ball-like grain in the red box. The grain was composed of a fibril structure that is similar to that of N3 and PM6 film. In the L8-BO film, a similar fibril structure can also be found on the edge of the flakes. In Supplementary Fig. 8 we put AFM images of PM6 and blends coated from room temperature to 100 °C. Despite grain of fibril size variation, similar fibril structure was observed in both pristine PM6 films and blend films across all temperature conditions. This could be evidence of the existence of PM6 fibril in blend films across all coating temperatures we investigated. Combining with the invariable spectrum of donor peaks during the film-forming process, we adopted the hypnosis that donor polymer shared similar packing attributes in both pristine films and blend films in GIWAXS analysis.

Moreover, the gradually shrinking grain size of Y6-based films in Supplementary Fig. 8 as temperature rises could match up with the improving $J_{sc}$ of Y6-based devices coated at the higher temperature, which could indicate the increasing D/A surface. As the coating temperature reaches 100 °C, similar fibril morphology is observed across three different films. This could be the evidence of the critical role of fibril structure in high-performance OSC devices.

GIWAXS images of pristine PM6 film and blend films were shown in Fig. 3e–h, and the corresponding in-plane (IP) direction and out-of-plane (OOP) direction line graphs are shown in Fig. 3i. Both pristine PM6 film and blend films take a face-on preferred packing motif, while N3's blend film also possesses edge-on signal in IP direction. From Fig. 3g, i, we found that PM6:N3 film possesses the most obvious scattering rings, which is the other evidence of the presence of crystal grains in PM6:N3 films. We first extracted packing attributes of pristine PM6 film and the results are shown in the first row of Fig. 3j, k. It turns out that PM6 has two periodicities in the OOP direction (~1.21 Å$^{-1}$ and ~1.67 Å$^{-1}$) and one major periodicity in IP direction (~0.285 Å$^{-1}$). By

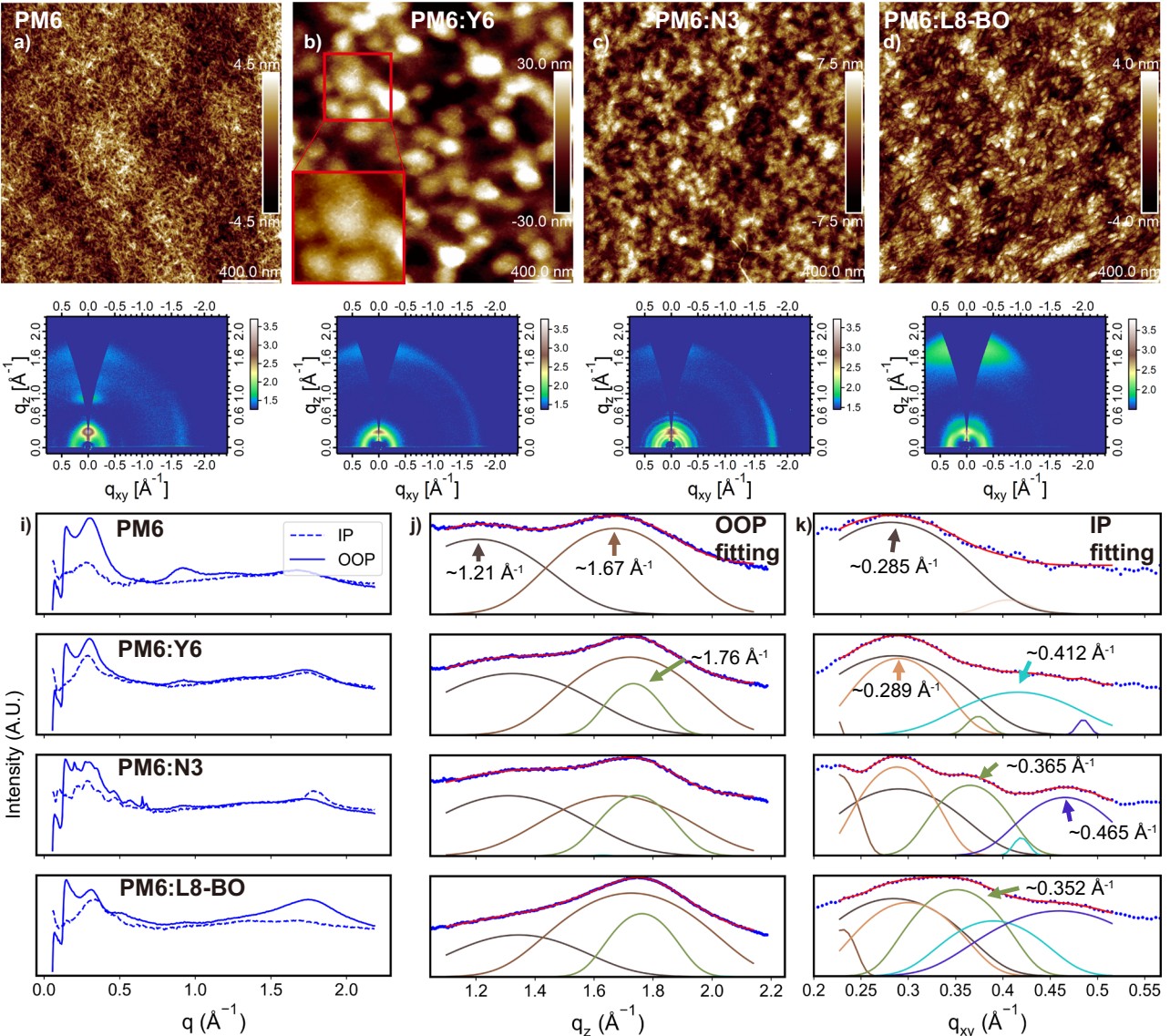

**Fig. 3 | AFM images and GIWAXS plots for pristine PM6 film and blend films.** AFM and 2D-GIWAXS plot for PM6 film: (**a**) and (**e**), PM6:Y6 film: (**b**) and (**f**), PM6:N3 film: (**c**) and (**g**), and PM6:L8-BO film: (**d**) and (**h**). 1D GIWAXS plot and peak splitting results are shown in **i–k**.

extracting both peak positions and full-width-at-half-maxima (FWHM) information of PM6 films and applying them in the fitting of blend films, we split GIWAXS peaks and results are shown in the lower section of Fig. 3j, k. We also performed GIWAXS characterization for blends coated under various temperatures, and the results are shown in Supplementary Fig. 9. Details of the GIWAXS fitting can be found in Supplementary Note 2.

In the OOP direction, all three Y6 analogs share a similar periodicity of ~1.76 Å⁻¹, which correspondence to 3.57 Å of $\pi$–$\pi$ packing distance. Such packing is a bit tighter compared with that of PM6 with a packing distance of 3.76 Å. In the IP direction, four major packing peaks were identified ranging from 0.25 Å⁻¹ to 0.5 Å⁻¹, but their relative intensities are varied. In the Y6 blend, two major peaks centered at ~0.290 Å⁻¹ and ~0.416 Å⁻¹ were identified, corresponding to periodicities of 21.66 Å and 15.10 Å. While in N3's blend, apart from the same 0.289 Å⁻¹ peak as Y6, two extra peaks centered at ~0.365 Å⁻¹ and ~0.466 Å⁻¹ were presented, corresponding to periodicities 17.21 Å and 13.48 Å. For L8-BO's case, the 0.365 Å⁻¹ peak once presented in N3 slightly shifted to ~0.351 Å⁻¹ (17.89 Å) while other peaks remained nearly the same as that presented in Y6 and N3. Since all three acceptor molecules share the same backbone and N3 has the lowest packing

hindrance, it is natural that N3 could have most of the packing styles that exist in Y6 and L8-BO. But the difference in packing preference in Y6 and L8-BO is intriguing: Y6's main packing peak is centered at ~0.289 Å⁻¹, which L8-BO also has but inferior compared with ~0.351 Å⁻¹ peak.

We applied the same analysis on GIWAXS results of films coated under various temperatures, the result of which is shown in Supplementary Fig. 10, yielding similar results as discussed above. The discussed difference in scattering peaks could imply the great difference in packing styles, which involves several acceptor molecules. However, referring dimer packing motif directly from the IP signal of GIWAXS is difficult due to the isotropic nature of our film.

The influence of such packing attributes on device performance is significant. In Supplementary Fig. 11 we analyzed and plotted GIWAXS results of the best spin-coated devices, which, qualitatively speaking, showed a heavy correlation between stacking change and performance deviance between small-area devices and large-area devices, where N3 systems showed the largest difference in packing styles and device performance, Y6 and L8-BO system changes less and thus retained most of their performance in small-area devices.

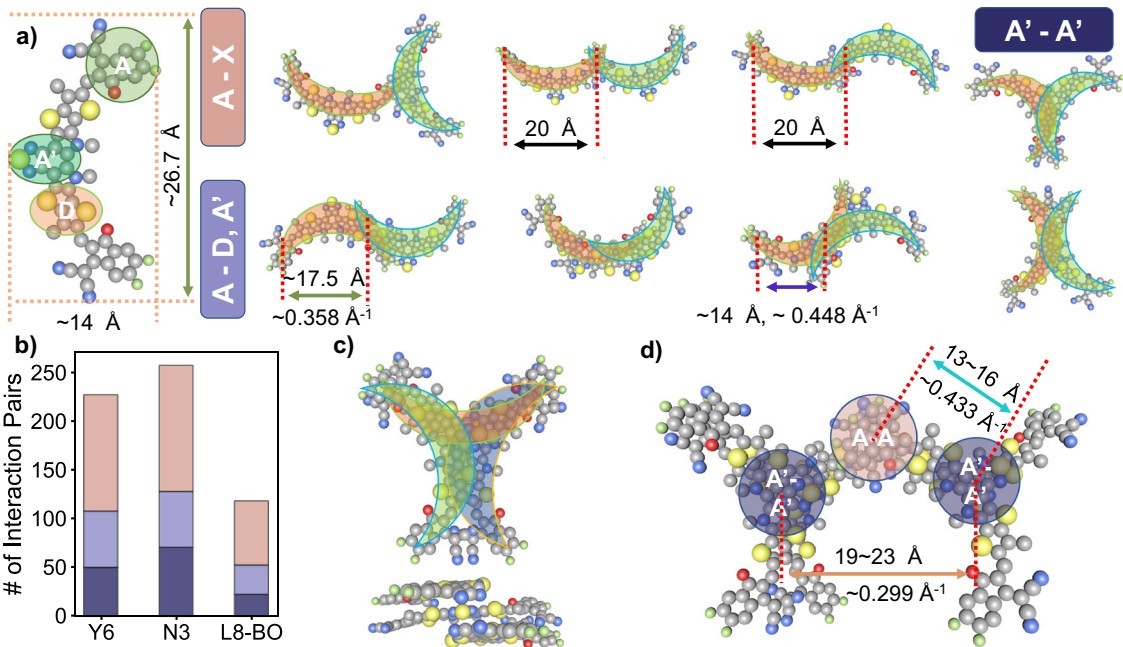

**Fig. 4 | Extracted packing dimers, trimers, and packing statistics of molecular dynamic simulation of Y6 analogs. a** Geometry of Y6 backbone and extracted dimer packing motif and measured periodicity, classified by interacting subgroups of two acceptor molecules. **b** Statistic of such classified dimer pairs extracted from molecular dynamic simulation. **c** Top and side view of star-shaped trimer extracted from Y6 and N3's molecular dynamic trajectory. **d** Extending trimer configuration extracted from Y6's trajectory.

## Molecular dynamic simulation of Y6 analogs

Molecular dynamic simulations were carried out to help investigate the influence of alkyl chains on the molecular packing of Y6 analogs and to understand GIWAXS signals. We systematically analyzed the balanced trajectory of three Y6 analogs and extracted eight types of dimers. The packing style of these dimers is shown in Fig. 4a. Packing style of these dimers is in accordance with reports elsewhere[28,29]. According to the different interacting sites of the dimer, we classified these eight types of dimers into three groups, and a statistical number of these three groups of dimers is shown in Fig. 4b. Details of classification and statistics can be found in Supplementary Note 3.

We separate the backbone of Y6 into three groups: terminal electron withdrawing group (A), center electron withdrawing group (A′), and electron donating fused rings in between (D). If only one subgroup of each molecule in a dimer pair forms π−π interaction with its counterpart, we classify it as A-X type interaction. Such interaction happens between one terminal group of a molecule and any other subgroup in another molecule, resulting T-shape, W-shape, and sparse S-shape dimer as depicted in Fig. 4a. If the π−π stacking involves more than one subgroup of a molecule, we further classify it depending on actual interacting types. If the interaction happens between the A group of a molecule and the D/A′ group of the other, such dimers will result in two types of closely packed S-shape dimer, and a C-type dimer, named A-D, A′ group. If the A′ group of a molecule is closely interacting with the A′ group of its counterpart, Y-shape and X-shape dimers would appear, and such dimers are classified in A′-A′ group. One would easily find that overlapping of the molecule backbone in a dimer follows A′-A′ > A-D/A′ > A-X, and so does the repulsion bring by alkyl chains. But larger overlapping of molecule backbone generally means higher electron delocalization and thus lower total electron energy, and one would expect a balance between electron potential and configurational potential of alkyl chains. Figure 4b depicts the result of such balance. The total number of interaction sites we found in a balanced trajectory follows N3 > Y6 > L8-BO, which agrees with our perception that N3 possesses lowest alkyl chain repulsion, followed

by Y6, and L8-BO. In Fig. 4a we further picked out chain-forming dimers and measured in-plane distance between π − π interacting sites. For the A-X group, a periodicity around 20 Å (corresponding to ~0.314 Å⁻¹) is anticipated, which might be the source of the peak centered at 0.290 Å⁻¹. In A-D, A′ group, two periodicities of ~17.5 Å (0.358 Å⁻¹) and ~14 Å (0.448 Å⁻¹) were found and might correspondent to the observed GIWAXS patterns at similar periodicity. So far, we could still not firmly assign packing of dimers to GIWAXS peaks, but with the evidence below we could make a better judgment.

If we enlarge our vision to microaggregates like trimer and beyond, one astonishing microcluster was found. Depicted in Fig. 4c, a star-shape trimer composed of two Y-shape dimers, which share one acceptor molecule in the middle, was found in both Y6's trajectory and N3's trajectory. These star-shaped trimers are nearly identical in their three arms, though the third arm (the one depicted in the lower section of Fig. 4c) is not as closely packed as the other two. The three arms could form identical A-X type dimers with surrounding acceptors and form a microstructure, for which we depict one typical example from MD trajectory in Fig. 4d. In this microaggregate, an acceptor molecule first interacts with acceptors from the trimer, forming an A-A type interaction, while the second acceptor closely interacts with the first one and assembles into another Y-shape dimer. We could easily identify two different types of π − π stacking sites in this microcluster: A-A site and A′-A′ site. The distance between an A′-A′ site and an A-A site would be slightly larger than half the length of a Y6 backbone, i.e., 13 − 16 Å (corresponding to a median periodicity of 0.433 Å⁻¹), while for the distance between two A′-A′ site will depend on the relative layout of two trimers, roughly around 19 Å ~ 23 Å (0.299 Å⁻¹), which could match up with 0.289 Å⁻¹ and 0.412 Å⁻¹ peak observed in GIWAXS.

## Assembly-controlled film-forming kinetics

Summarizing the experimental and simulation result we've presented so far, several key factors are to be considered comprehensively here: (1) in-situ spectroscopy suggests a grow-dominated pathway for N3 and Y6 blends, which correspond to an increment of microassemblies

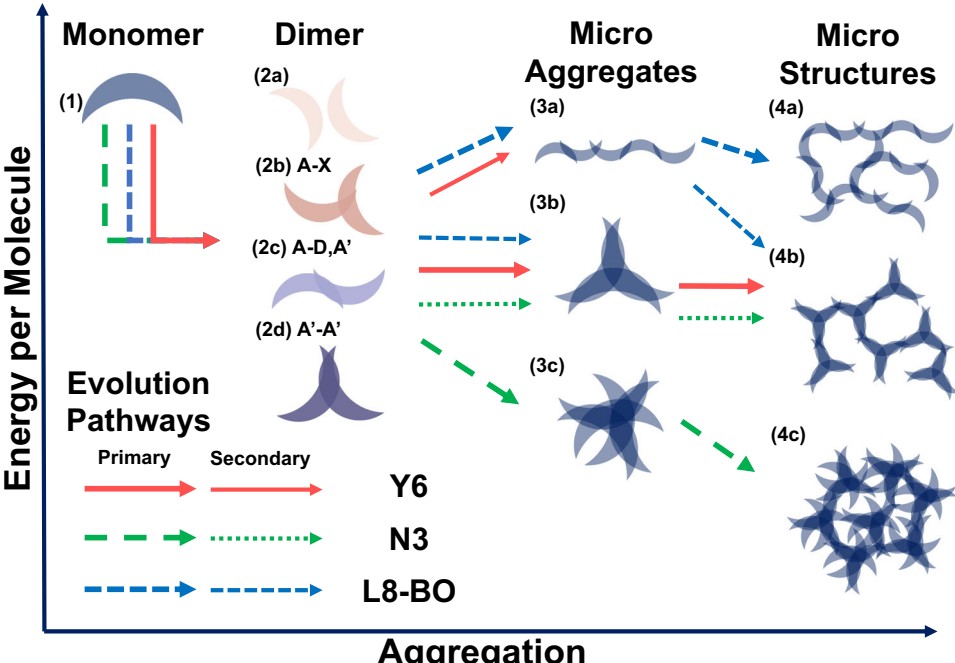

**Fig. 5 | Schematic illustration of assembly-controlled film-forming dynamic of Y6 analogs.** The assembly of molecules during a slot-die coating session happens stage by stage. The phases of molecule assembly, i.e., aggregation stages were laid horizontally. In each stage, configurations with lower electron configuration energy were laid at the lower position of the group. Starting from the same monomer (1) in upper left, the synergetic effect of electron delocalization and sidechain repulsion creates different populations of dimers in Y6, N3, and L8-BO systems, which we further classifies into weakly interacting(2a) dimers, A-X type interacting(2b) dimers, A-D,A' type interacting dimers, and A'-A' type interacting(2d) dimers. Following the arrows of different shapes, L8-BO mainly aggregates into chain like(3a) micro aggregates, gradually red shifts absorbing edge and finally creates fibril structure(4a) as observed in AFM characterization. Y6 mainly forms star-shaped trimer(3b), depicting the growing pattern in in-situ UV-Vis spectroscopy and forms the grain-like structure in AFM images. When these star-shaped dimers linked together and form network structure(4b), characteristic GIWAXS pattern was formed. While for N3, most dimers are in 2d type and forms dense aggregation(3c), which further grows into oversized crystals(4c), leading to the catastrophically poor device performance.

that have a fixed absorption characteristic. (2) Two dominating packing motifs exist in the in-plane direction of the Y6 blend film (0.289 Å$^{-1}$ and 0.412 Å$^{-1}$ separately). (3) Microstructures formed by near-unified star-shape trimers which possess periodicity matched well with GIWAXS patterns are presented in Y6 and N3 simulation trajectories. By reviewing these findings, it would be so natural to come up with an assembly-controlled film-forming kinetics in which the star-shaped trimer answered the key question to spectral evolution and GIWAXS feature: In blending Y6 assembles into a large quantity of star-shaped trimer, leading to the increment of characteristic absorption centered around 847 nm. While in dried films, closely interacting star-shape trimers assemble by overlapping the terminal groups, creating the iconic periodicity as depicted in Fig. 4d. We consider such kinetic highly convincing, not only did it successfully explain the evolution of spectrum for both Y6 blend and pristine film but also have solid GIWAXS results and simulation results as back support. Furthermore, such assembly-controlled film-forming kinetics can be extended to explain N3 and L8-BO systems, which will be further elucidated below.

We illustrate acceptors in different phases of aggregation in Fig. 5. Configurations of acceptor backbones with higher electron energy were put in higher positions, while those has lower energy go to lower places. Following our analysis in the former section, two acceptor molecules interacting with each other would result in A-X type dimer(2b), A-D, A' type dimer(2c) and A'-A' dimer(2d), or less-locked-but-still-interacting 2a type dimer. Naturally, acceptor molecules would prefer configurations at lower positions to minimize their electron energy, but such a tendency will be modulated by the repulsion of sidechains.

We will demonstrate such modulation by a case study of N3, Y6, and L8-BO separately.

(1) N3's case: the repulsion of sidechains does not severely affect the assembly of the backbones. Following the green arrow depicted in Fig. 5, most of the N3 molecules form 2d dimers and further assemble into 3c microaggregates, leading to the H-aggregation feature observed in N3's UV-vis spectrum in stage one. Later, these 3c type microaggregates swiftly assemble into 4c microstructure and crystallize, raising the baseline of the UV-vis spectrum and creating multiple scattering peaks in GIWAXS. The residual N3 molecules in low concentration assemble into 2c and 2d dimers and further compete in forming 3a and 3b microaggregates, creating the competing growing and moving kinetic observed in-situ spectroscopy.

(2) Y6's case: moderate repulsion of sidechains decreases the popularity of 2d dimers and eliminates the possibility of them assembling into 3c microaggregates. As depicted by red arrows in Fig. 5, 3a microaggregates start to appear and leading to the J-aggregation feature we observed. On entering stage two, as a consequence of energy favorability massive 3b trimer was generated, leading to growing dominated kinetic in in-situ spectroscopy. Further, the assembly of 3b microaggregates into 4b microstructure creates the iconic feature in GIWAXS as observed.

(3) L8-BO's case: large repulsion of sidechains further inhibits the formation of 3b trimer, shifting the assembly balance toward chain-like 3a type microaggregates. As depicted by blue arrows in Fig. 5, the majority of L8-BO molecules assemble into 3a microaggregates. But the repulsion is not completely blocking the formation of star shape trimer and interwinding chains, and a small amount of L8-BO still assembles into 3b trimer, together with intertwined chains in the film it creates a similar GIWAXS

feature as Y6 had, but the majority feature yields from the chain-like structure.

The kinetic can also explain the different evolving behavior of acceptors in blends and pristine solution. We compared the parking position of moving peaks of blend and pristine films in Supplementary Fig. 12a–c as we have discussed in analyzing the result of in-situ spectroscopy. The introduction of PM6 generally inhibits the assembly of Y6 analogs, but the assembly of A'-A' type packing was more severely affected. This is anticipated since absorbed Y6 analogs on PM6 fibrils' surface would greatly reduce the availability of its A' group while terminal groups are less affected. Such a result encourages Y6 and N3 to take moving kinetic, shifting the parking position of the moving peak toward a longer wavelength. For L8-BO's case, the introduction of PM6 overall inhibited the further assembly of conjugated chain structure, resulting in a blue shift of the moving peak position. When the concentration of Y6 analogs reduced, the chance of two acceptors colliding and interacting was reduced, forcing the balance of dimer assembly drifts toward the 2a and 2b type. For N3's case, over-aggregation in the early phase of solvent evaporation results in a low concentration of acceptor in the solution, greatly shifting the kinetics in favor of moving kinetic. For clarity, we put the further discussion of our kinetic model in Supplementary Fig. 12 and the topic about temperature, D/A ratio, and solution concentration in Supplementary Note 4, from where we further verified the capability of our kinetic model by explaining how external effects affect the assembly, and hierarchical structural evolution thereafter. We further tried coating blends on glass substrates with a substrate temperature of 115 °C, the results of which are shown in Supplementary Fig. 13. After fitting with our grow-move two peak model the portion of growing peaks have all increased, which is in accordance with our kinetics, further strengthening the liability of our kinetic model.

From AFM images taken at different temperature and related device performance, we could see how this assembly kinetic affected microstructures hierarchically. The star-shaped microaggregate could further interact with each other, winding up donor fibrils and forming yarn-ball-like structures, which greatly reduces surfaces available for charge separation, leading to poor $J_{sc}$ and device performance in low-temperature-coated Y6-based devices. If the aggregation gets stronger and happens in the earlier stage of solvent evaporation, large crystals will form like that observed in N3's case, which is undesirable in an OPV device. If one maintains the assembly of acceptors in a continuous style like that in L8-BO's case, a temperature insensitive film can form, and higher PCE is expected.

Moreover, we performed the same UV-vis spectroscopy analysis on the same systems with toluene as solvent. As shown in Supplementary Fig. 14 and Supplementary Fig. 15, it is astonishing to find that compared with o-xylene, toluene tends to facilitate aggregation of acceptors and thus leads to a higher portion of growing kinetics and crystallization. Details of the analysis can be found in Supplementary Note 5. The result is anticipated in our proposed kinetic model since toluene poses lower resistance packing with acceptor backbones compared with o-xylene, thus acting as a better lubricant and helping better in the assembly of acceptors.

Following the analysis of our kinetics, it is now possible to discuss a bit more in detail about the performance deviation between small-area and large-area devices. Our kinetic model indicates an absorbing behavior of acceptor molecules onto the donor fibrils, which is generally bad for spacing between donor and acceptor and bring extra energy loss to the system[30]. When focused on Y6 and L8-BO systems, we find out that loss in $V_{oc}$ when scaling up for L8-BO systems is nearly doubled compared with Y6 systems (0.048 V vs 0.027 V). We attribute such extra loss to the packing style shift of L8-BO systems. From Figs. 3 and S15 we could tell that L8-BO tends to assemble in a more relaxed formations when being slot-die coated (smaller **q** vector peaks

dominate compared with spin-coated films), which could induce more coupling between charge transfer state and molecule vibronic states and thus brought excess energy loss[31].

## Discussion

We managed to precisely describe the film-forming kinetics on a sub-molecular level. This very detailed kinetics could strengthen our capability in the manipulation of nanoscale morphology much stronger than before, such as targeted induction or inhibition of molecular assembly either by intentional molecular design or selective addition of certain additives. By understanding the primary assembly of acceptor molecules, additive molecules that target certain binding sites can be designed and synthesized to bind with acceptors, encouraging or discouraging certain types of assembly, and guiding the assembly toward another direction. For example, suppose a conjugated solid additive can be synthesized to target the core acceptor unit of Y6 and N3 molecule; such binding will reduce the formation of the star-shaped trimer, thus forcing the kinetics to move toward moving kinetic. Moreover, the host-guest strategy might also be applied to facilitate more favorable assembly patterns and multiscale morphology in the formation of active layers. Furthermore, our experience here provides tools and examples on how to characterize such assembly in the field of organic photovoltaics. The kinetic model we proposed here can surely be one of the base rocks toward the rational optimization of device fabrication parameters.

Combining various characterization techniques and simulation has proven effective in answering the complex questions in the field of organic photovoltaics. First of all, the in-situ UV-vis spectrum is an excellent tool for monitoring the kinetics of the film-forming process, so long as one could extract similar characteristic peaks from collected spectrums. In this work, the evolution of major absorption peak of acceptors demonstrated the existence of both continuous extension kinetic and fixed growth kinetic. Secondly, with a combination of AFM and GIWAXS characterization, we could narrow down the scale of structure to be further investigated and locate the key difference in the scale of tens of angstroms. Then, thanks to molecular dynamic simulations, we could extract critical dimer and trimer configurations which helped explain GIWAXS results. Comprehensive results finally unveiled the last uncertainty in the structure of microaggregate and fully explained both spectral evolution and GIWAXS characterization. Our method reported here could be a paradigm in the research of packing structures of non-crystalline thin films, from solely the structural details to the kinetics of their formation.

We believe assembly behavior on various scales, from dimers to microstructures, determines the film morphology hierarchically, and photonic physic happens within. Though primitive, the reported kinetics could be an inspiring glance into the film-forming procedure of solution-processed organic electronics. If characterization techniques could be utilized to extract multiscale structural information from an organic thin film even during the film-forming procedure, both our understanding of the structure-performance relationship and commercialization of organic photovoltaics would have one great step forward.

## Methods

### Materials

PM6 used in this work was purchased from eFlexPV Limited (China), and Y6, N3, and L8-BO were obtained from Hyper, Inc. (China). The solvent o-xylene was purchased from Sigma-Aldrich. PEDOT:PSS(Al 4083) used to prepare GIWAXS and AFM samples were purchased from Heraeus. The PET substrate with etched ITO and glass substrate with etched ITO was purchased from Youxuan New Energy Technology Co., Ltd. The PET substrate with printed silver grid used for manufacturing large-area devices was purchased from Suzhou NanoGrid Technology Co., Ltd.

## Slot-die process and in-situ UV-vis spectroscopy

The slot-die process was conducted on a lab-assembled slot-die apparatus, with an integrated Filmetrics F-20 spectrometer. All coating sessions were conducted with a coating speed of 45 mm/s and solvent injection speed of 5 µL/s. The strip width of slot-die coating is 10 mm, and the gap between the slot-die head and the substrate is controlled at 125 ± 10 µm by using a contact sensor to calibrate servo motors. Transmission mode is used in the collection of in-situ spectroscopy signal.

Solutions with a concentration of 16 mg/mL and a mass ratio of D:A = 1:1.2 were used to experiment. Acceptor solutions with a concentration of 8.73 mg/mL are used to coat the pristine films of acceptors. Later, blends with D:A = 3:1, concentrations of 16 mg/mL and 4 mg/mL were used to investigate the influence of donor and concentration on kinetics. Acceptor solutions of 4 mg/mL and 1 mg/mL were also used.

Absorption baseline was taken with PET substrate before each coating session and the coating starts after the solution from the slot-die head had made contact with the substrate. The UV-vis spectrum was taken at around 80 mm position out of a total 130 mm coating stripe to avoid the beginning and end phases of the coating session. The integration time of the spectrometer was set to 7 ms, starting from the same time when the slot-die head moves and ends until no significant evolution can be observed in the real-time spectrum. In-situ UV-vis data was later processed using Python and fitted according to Supplementary Note 1.

## GIWAXS and AFM characterization

A layer of PEDOT:PSS (Al 4083, 1:1 v/v in iso-propanol) was first coated onto PET substrate, with a coating speed of 13 mm/s and solvent injection speed of 3.3 µL/s (200 µL/min). The gap between the slot-die head and the substrate is controlled at 125 ± 10 µm, while the temperature of the coating plate is set at 40 °C. The film was later annealed at 90 °C for 10 min before further operation.

The blend solution was later coated onto PEDOT film using the same parameters as in the in-situ UV-vis spectroscopy coating session and was cut into 10 mm × 10 mm squares. Deionized water was used to dissolve PEDOT:PSS layer and let blend film afloat. The film was later transferred to a glass substrate for AFM characterization, and to silicon substrate for GIWAXS characterization.

AFM characterization was conducted using Bruker Multimode 8HR in Scan Asyst mode in the air, and GIWAXS analysis was conducted by the XEUSS SAXS/WAXS system.

## Molecular dynamic simulations

The MD simulation of pristine Y6, N3, and L8-BO was performed in LAMMPS package[32]. The OPLS-AA force field was used to describe the bonded and non-bonded potentials[33]. The equilibrium bond lengths, equilibrium bond angles, and dihedrals were reparametrized using DFT calculation at ωB97XD/6-31 G(d,p) level. The partial charge of atoms was fitted with the restrained electrostatic potential charge (RESP charge) method using the Multiwfn[34] program.

For each simulation, 200 acceptor molecules were used. Molecules with random position and orientation were placed into a large cubic box with periodicity boundary conditions at low density. First of all, NPT-MD was performed for 30 ns at 650 K to equilibrate the structure. The box was then cooled to 300 K at a rate of 10 K/ns. Finally, a second NPT-MD was performed at 300 K for 30 ns to obtain the balanced structure.

All of the MD simulations employed a velocity-Verlet integrator at 2.0 fs time step with SHAKE algorithm. The temperature and pressure were controlled using a Nose-Hoover thermostat and barostat respectively. The cutoff for non-bonded interactions was set to 12 Å and the electrostatics was solved using the particle–particle particle-mesh method. All of the results were analyzed from the last 10 ns of the MD simulations and averaged from three independent boxes.

## Fabrication of large area devices and measurement of device efficiency

Silver-grid patterned on PET substrates (1.5 Ω/sq, T ≥ 85%) was selected as the bottom electrode. The same apparatus used to perform in-situ UV-Vis spectroscopy characterization was used to fabricate large area device. The inverted solar cell devices were processed in an ambient atmosphere, using the PET silver-grid/PH1000/ZnO/active layer/MoOx/Ag structure. The PEDOT:PSS (Clevios PH1000) was diluted with isopropyl alcohol at a ratio of 1:4 v/v, and then slot-die coated on the PET/Ag-grid substrates with an injection speed of 3.5 µL/s and coating speed of 10 mm/s. The hybrid electrodes were baked at 120 °C for 20 min in ambient air. The ZnO layer was coated by slot-die coating ZnO sol-gel with isopropyl alcohol solvent (15 mg/mL), injected at a speed of 1.2 µL/s with a coating speed of 10 mm/s. The active layer solutions were prepared in o-xylene with total concentration of 16 mg/mL and the ratio of donor:acceptor was maintained at 1:1.2 w/w. The active layers were slot-die coated with an injection speed of 1.7 uL/s and coating speed of 13 mm/s in ambient conditions, after heating plate reaches the designated coating temperature. In all coating sessions, the gap between slot-die head and the substrate is controlled at 125 ± 10 m by using a contact sensor to calibrate servo motors. The strip width of slot-die head is 13 mm. The samples were then transferred into a vacuum evaporation chamber to deposit the top electrode. At a vacuum level of ≈1.0 × 10⁻⁶ mbar, a thin layer of MoOx (5 nm) and Ag (160 nm) were thermally evaporated onto the samples with different shaped templates.

The J-V characteristics of the devices were assessed under AM 1.5 G (100 mW/cm) with a SAN-EI ELECTRIC XES-1004SE-200S solar simulator. Light intensity was calibrated with a Newport Oriel 91150 VKG5 Si-based solar cell and a Newport Oriel PN 91150V Si-based solar cell. *J–V* characteristics were recorded with a Keithley 2400 source meter unit. Tested area of devices was controlled to 1.036 cm² with a mask.

## Fabrication of small area devices and measurement of device efficiency

The rigid small-area devices were fabricated by spin coating with an inverted device structure of glass/ITO/ZnO/active layer/MoOx/Ag. The patterned ITO glass was cleaned by sequential sonication in soap with deionized (DI) water, then in DI water, ethanol, and finally in isopropyl alcohol for 30 min each. After ultraviolet-ozone (Ultraviolet Ozone Cleaner, Jelight Company, USA) treatment for 15 min, the ZnO precursor was spin-coated on the ITO substrate at 4000 rpm to form an electron transporting layer ZnO. Then, the substrate was baked at 200 °C for 30 min. The Donor and acceptor materials were dissolved in an o-xylene solvent with a total concentration of 16.7 mg/ml (D:A = 1:1.2 w/w). The active layers were spin-coated and post-processed in an N₂ glove box at room temperature following conditions listed in Supplementary Table 2. At a vacuum level of ≈1.0 × 10⁻⁶ mbar, a thin layer (5 nm) of MoOx was deposited as the anode interlayer. Finally, a 160 nm of Ag was deposited onto the active layer to form a back electrode. Photovoltaic performance of small-area devices were measured in a N₂-filled glovebox. Newport Thermal Oriel 91159 A solar simulator was used for *J-V* curves measurement under AM 1.5 G, and the light intensity was calibrated with Newport Oriel PN 91150 V Si-based solar cell. Typical cells have device areas of ~4 mm². A mask with well-defined area (2.56 mm²) was used in *J–V* characteristics as well. *J–V* measurement signals were recorded by Keithley 2400 source-measure unit.

**Reporting summary**

Further information on research design is available in the Nature Portfolio Reporting Summary linked to this article.

## Data availability

Datasets generated and used for the analysis of molecular dynamic trajectories are stored in the GitHub repository[35], which can be found at https://github.com/PulsEternal/PackingAnalysisY6analogues. All fitting results of conducted in-situ UV-vis spectroscopy experiments, which are also the raw data used to generate the statistical plot of Supplementary Fig. 12 can be found in Supplementary Data 1. Other data that support the findings of this study are available from the corresponding author upon request.

## Code availability

Code used to generate feature vector used in the analysis of molecular dynamic trajectories are stored at the GitHub repository[35], which can be found at https://github.com/PulsEternal/PackingAnalysisY6analogues.

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

## Acknowledgements

This work was financially supported by the National Natural Science Foundation of China (NSFC) (Nos. 52373177, 52073068, 22135001, and 21721002) and the Strategic Priority Research Program of the Chinese Academy of Sciences (No. XDB36000000).

## Author contributions

H.Z. and C.T. contributed equally to this article. H.Z. and J.Z. designed the experiments and wrote the paper. H.Z. and L.Z. designed, performed, and analyzed molecular dynamic simulations. H.Z. and C.T. designed and assembled the integrated slot-die in-situ UV-Vis spectroscopy apparatus, fabricated the large-area devices, and performed in-situ spectroscopy and device performance test. H.Z. and J.Z. performed GIWAXS characterization and analyzed results. H.Z., Z.Z., C.T., and M.X. performed AFM characterization. Z.W. supervised the project and wrote the paper.

## Competing interests

The authors declare no competing interests.
