## [Peer Review File · Nature Communications]

Concretized Structural Evolution Supported
Assembly-Controlled Film-Forming Kinetics in Slot-die Coated
Organic PhotovoltaicsREVIEWER COMMENTS

Reviewer #1 (Remarks to the Author):

The manuscript concretized the structural evolution during film-forming kinetics of slot-die coating on a sub-molecular level by combining in-situ UV-vis spectroscopy, AFM, GIWAXS, and molecular dynamics simulation. It is crucial that combining various characterization techniques and simulation is proven effective in answering the complex questions of the structural-performance relationship in the field of organic photovoltaics. The proposed assembly-controlled film-forming kinetics would contribute to screening large-area device fabrication conditions and inspire the future design of novel photovoltaic materials. Thus, I support the publication of this work in Nature Communications. Before publishing the manuscript, the comments below should be addressed.

1. In Fig. 2e, the lower slope of the purple line means that the PM6:N3 blend film has obvious tailing at the edge of the final state absorption. Does this tailing mean that the N3 stacking is more disordered?
2. The use of $\lambda(0-0)$ is inappropriate in describing the evolution of the spectrum. Per the authors' analysis, there are different absorbing types in different evolution pathways, i.e., the growing pattern and moving pattern. Multiple electron transitions are involved in the process, and thus it is blurring to assign $\lambda(0-0)$ peak to a certain transition type.
3. In Fig. 2d-f, the intensity of the acceptor peak in PM6:N3 blend film is lower than that of the donor peak at the beginning, but there is no such phenomenon compared with the other two systems. Please give more explanation.

The following are some format issues.

4. In Fig. S6, the scale bar is missing in AFM height graphs. Authors need to supplement.
5. Fig. S8 is closely related to the discussion of GIWAXS characterization in the manuscript but hasn't been mentioned. Authors need to check that the images in SI are all explicitly cited in the manuscript.

Reviewer #2 (Remarks to the Author):

The manuscript titled "Concretized Structural Evolution Supported Assembly-Controlled Film-Forming Kinetics in Slot-die Coated Organic Photovoltaics" by Zhang et al. investigated the correlation between the morphology of the active layer and the performance of organic photovoltaic (OPV) devices produced using the slot die coating method. Through the application of in situ spectroscopic analysis, the study revealed significant effects of the side-chain structure of acceptor materials on the film formation

process of the active layer. The utilization of AFM and GIWAXS measurements provides clear insights into the formation of distinct microstructures in the active layers resulting from the use of acceptors with different side-chains. Additionally, molecular dynamics simulations are employed to elucidate the mechanism by which the molecular structure of acceptor materials influences the film formation process of the active layer. The research findings contribute to the advancement of high-performance organic acceptor materials and the optimization of the morphology of the organic active layer, thereby offering substantial potential for achieving breakthroughs in the performance of OPV devices. Therefore, I recommend accepting the manuscript for publication in Nat. Commun. Please find my detailed comments listed below.

1. The study presented in this paper provides a detailed investigation of morphology evolution. However, the relationship between this morphological difference and device performance could be further explored. For instance, it is observed that devices prepared by slot-die coating exhibit generally lower Voc compared to those prepared by spin coating. This raises the question of whether there exist other microstructures, beyond the dimer structures and morphologies discussed in the article, that are more favorable for enhancing OPV performance. The authors are suggested to provide some brief discussions on this topic.
2. A more detailed description of the in-situ measurement of the absorption spectra would be appreciated. Specifically, it would be beneficial if the authors could clarify whether the spectrum is measured in reflection mode or transmission mode. Furthermore, it is important for the authors to explain how the absorption spectrum is determined, whether it is calculated using the equation $A=1-R$, $A=1-T$, or $A=1-R-T$. Additionally, it would be helpful to specify whether the absorption spectrum is measured on devices or using organic blend films.
3. In this study, the authors summarize the structural formation process of active thin films containing acceptors with different side chains deposited via xylene. It would be beneficial for the authors to clarify whether the observed patterns of structural formation can also be applicable to blended films deposited using other solvents.
4. A higher coating temperature appears to improve device performance. I wonder if the authors could further increase the coating temperature to over 100°C to investigate whether significantly altering the coating conditions can lead to a change in the growth mode of the blended active layer.
5. It appears that the coating temperature affects both Jsc and FF of the solar cells based on Y6 and L8-BO, but has minimal impact on the FF of the solar cell based on N3, despite significant changes in the morphology of the active layer. It would be helpful if the authors could provide an explanation for this observation.

6. It is recommended to provide JV curves and EQE spectra for all the solar cells investigated in this study. This would enable a better understanding of the role played by the acceptor side chain in determining the morphology and performance of slot-die coated solar cells.

7. In the first paragraph of the results section, the statement "Later, AFM and GIWAXS results indicate large morphology differences in tens of Angstroms and larger scales" is unclear and requires rephrasing. It is unclear what is meant by "morphology differences in tens of Angstroms."

8. In the second paragraph of the results section, it is mentioned that "PCEs of 1 cm² devices made from PM6/Y6 analogues are dramatically different in numbers and temperature dependency". The temperature dependency mentioned here needs to be further clarified. It should be specified that this temperature refers to the substrate temperature rather than the temperature of the slot-die head or any post-annealing temperature.

9. In the first paragraph of the discussion section, it is stated that "our capability in the manipulation of nanoscale morphology much stronger than before, such as targeted induction or inhibition of molecular assembly either by intentional molecular design or selective addition of certain additives." The authors should provide further clarification on how the results of this study contribute to the improvement of the selection rule for additives.

Reviewer #3 (Remarks to the Author):

In this manuscript, Zhang et al. did a fascinating work by incorporating multiple experimental procedures and simulations to unveil the critical assembly structure and film-forming kinetics of famous Y6 analogues in slot-die-coated large-area organic photovoltaics. Each of the used characterization techniques was broadly applied to the topic of organic photovoltaics, but the authors made an inspiring elucidation on how to combine them to yield more profound insights. Among these, the proposition of a star-shaped trimer is the most fascinating part of the work, where authors did their best to concretize the assembly structure, via both in-situ spectroscopy and GIWAXS characterization. The evidence presented in this manuscript is convincing enough to support the proposed structure and the following kinetic model. I recommend the manuscript be published in Nature Communications after minor revisions. Several concerns that could be addressed are listed as follows:

1. The presented UV-vis spectrums in Fig. 2 have significant differences compared with each other. But since the authors only showed one in-situ UV-vis spectrum for each condition (including that in supporting information), this raises the concern of repeatability of the observed phenomenon. It would

be much better if the author could provide extra information on the repeatability of UV-Vis spectroscopy.

2.The authors mentioned the formation of nanocrystals and their size dependence on coating temperature. But no other experimental evidence except for the baseline shift in UV-vis spectroscopy is provided. Further evidence is highly desirable if the authors would like to make this claim.

3.The AFM image of PM6 in Fig.3a is unprecedented in previous reports. Authors should perform the AFM characterization again to further investigate the origin of such a distinguished branch-like structure. Also, authors should add a color bar to the AFM images.

4.In Supplementary Note 3 and Fig.S11, authors brilliantly designed feature vectors for automated packing analysis of molecular dynamics trajectory. Such a tool is critical to the discovery of star-shaped trimers and other assembly forms from a large quantity of molecular dynamic trajectories. However, the definition and the calculation method of the principal axis seem to be missing from the supplementary note. The authors should further elucidate the procedure to calculate the principal axis, as well as the generation of mentioned feature vectors.

5.The analytical method used in the manuscript could be inspiring to the community. Moreover, the ML model used to automatically classify the packing motif in a molecular dynamics trajectory would be valuable in many more scenarios. We here suggest authors publicize the model and the generated dataset for further applications.

Response to the comments of reviewers

We would like to thank the referees for spending time on this paper and providing invaluable comments which substantially helped improve the quality of the paper. The manuscript has been revised according to the comments point-by-point.

Reviewer #1 (Remarks to the Author):

The manuscript concretized the structural evolution during film-forming kinetics of slot-die coating on a sub-molecular level by combining in-situ UV-vis spectroscopy, AFM, GIWAXS, and molecular dynamics simulation. It is crucial that combining various characterization techniques and simulation is proven effective in answering the complex questions of the structural-performance relationship in the field of organic photovoltaics. The proposed assembly-controlled film-forming kinetics would contribute to screening large-area device fabrication conditions and inspire the future design of novel photovoltaic materials. Thus, I support the publication of this work in Nature Communications. Before publishing the manuscript, the comments below should be addressed.

Response to comment: We are grateful to the reviewer for the positive comments on our work.

1. In Fig. 2e, the lower slope of the purple line means that the PM6:N3 blend film has obvious tailing at the edge of the final state absorption. Does this tailing mean that the N3 stacking is more disordered?

Response: We generally agree with the claim proposed by the reviewer. The absorption of an ordered stacking will be more unified and centralized, reducing the FWHM of the observed peak. However, if the packing is more disordered, the conjugation length in a film will vary to a larger extent than the orderly stacked film, creating a large variance in the transition energy of molecular orbital and thus a broader absorption peak. In our reported case, the obviously smaller slope of PM6:N3 blend came from a generally broader absorption peak. Thus, we could attribute the broadening of the absorption peak of PM6:N3 film to the more-closely-but-more-disordered stacking of N3 molecules. To address the concern, we have added the following brief discussion to our manuscript related to the discussion on page 6 as follows:

“And the lower slope of the absorption edge, i.e., the purple line in **Figure. 2e**, indicates a large variance in electron transition energy. The phenomenon suggests that PM6:N3 film may possess a more-closely-but-more-disordered stacking of acceptor molecules, further depicting its uniqueness compared with the other two Y6 analogues.”

2. The use of $\lambda(0-0)$ is inappropriate in describing the evolution of the spectrum. Per the authors' analysis, there are different absorbing types in different evolution pathways, i.e., the growing pattern and moving pattern. Multiple electron transitions are involved in the process, and thus it is blurring to assign $\lambda(0-0)$ peak to a certain transition type.

Response: We thank you for the careful reading and pointing out this misleading description. We agree that ascribing the absorption peaks to the electron transition between the ground state and the lowest first excited state is inappropriate if the molecule structure behind the electron transition varies. Following your advice, we changed our description of the acceptor absorption peak from “ λ_{0-0} peak” to general “major absorption peak” to avoid the mislead.

3. In Fig. 2d-f, the intensity of the acceptor peak in PM6:N3 blend film is lower than that of the donor peak at the beginning, but there is no such phenomenon compared with the other two systems. Please give more explanation.

Response: We thank you for addressing the abnormal phenomenon, which we failed to elucidate in detail. Multiple experiments have confirmed that the phenomenon, as shown in **Figure R3**, not being a result of an accident or mistake during our experiment procedures but connected to the attribute of the N3 molecule. The solution containing N3 molecules aggregates in an early stage of coating, prior to the observation window of our in-situ UV-vis spectroscopy. This reduces the concentration of N3 molecules in the solution, resulting in a lower peak position compared with the donor absorption peak.

To address the concern, we have added the following description and explanation to the end of the paragraph on page 7:

“Over crystallization and change in the ratio of donor and acceptor also reflects on the initial state of the absorption spectrum. Compared with PM6:Y6 and PM6:L8-BO, the acceptor absorbance peak in the PM6:N3 blend is obviously lower. Such phenomenon can also be

observed in spectrums coated under other temperatures.”

The following are some format issues.

4. In Fig. S6, the scale bar is missing in AFM height graphs. Authors need to supplement.

Response: Thank you for carefully reading our supplementary information and pointing out the missing scalebar in our AFM images. After our inspection, we found that all our AFM images lack a scalebar, and thus we reprocessed all our AFM figures, including **Figure.2 a)-d), Figure S6.**

5. Fig. S8 is closely related to the discussion of GIWAXS characterization in the manuscript but hasn't been mentioned. Authors need to check that the images in SI are all explicitly cited in the manuscript.

Response: Thank you for addressing your concern. We modified our discussion on the GIWAXS analysis and added the direct citation in the manuscript on page 13 as:

“We also performed GIWAXS characterization for blends coated under various temperatures, and the results are shown in **Figure S13**. The same fitting procedures are also applied, and the results are put in **Figure S14**. Details of the GIWAXS fitting can be found in **Supplementary Note 2.**”

Reviewer #2 (Remarks to the Author):

The manuscript titled "Concretized Structural Evolution Supported Assembly-Controlled Film-Forming Kinetics in Slot-die Coated Organic Photovoltaics" by Zhang et al. investigated the correlation between the morphology of the active layer and the performance of organic photovoltaic (OPV) devices produced using the slot die coating method. Through the application of in situ spectroscopic analysis, the study revealed significant effects of the side-chain structure of acceptor materials on the film formation process of the active layer. The utilization of AFM and GIWAXS measurements provides clear insights into the formation of distinct microstructures in the active layers resulting from the use of acceptors with different side-chains. Additionally,

molecular dynamics simulations are employed to elucidate the mechanism by which the molecular structure of acceptor materials influences the film formation process of the active layer. The research findings contribute to the advancement of high-performance organic acceptor materials and the optimization of the morphology of the organic active layer, thereby offering substantial potential for achieving breakthroughs in the performance of OPV devices. Therefore, I recommend accepting the manuscript for publication in Nat. Commun. Please find my detailed comments listed below.

Response to comment: We are grateful to the reviewer for the positive comments on our work and high thoughts on the manuscript.

1. The study presented in this paper provides a detailed investigation of morphology evolution. However, the relationship between this morphological difference and device performance could be further explored. For instance, it is observed that devices prepared by slot-die coating exhibit generally lower V_{oc} compared to those prepared by spin coating. This raises the question of whether there exist other microstructures, beyond the dimer structures and morphologies discussed in the article, that are more favorable for enhancing OPV performance. The authors are suggested to provide some brief discussions on this topic.

Response: Thank you for addressing your concern. We noticed that previous reports have pointed out several factors that could affect the open circuit voltage of OPV devices. Donor-acceptor spacing (Nat. Commun. 2021, 12, 6679), packing of acceptor molecules (Nat. Energy 2021, 6, 605-613), and even orientation of such packing will affect the V_{oc} of devices (ACS Energy Lett. 2019, 4, 1057-1064). Furthermore, previous reports have also pointed out the close coupling between molecular vibration and non-radiative energy loss (Adv. Energy Mater. 2018, 8, 1702227), which could also affect device V_{oc} .

To further understand the effect of molecule assembly and structure on the V_{oc} in our system, we fabricated small-area devices using o-xylene with spin-coating technique and performed GIWAXS analysis to investigate the difference in microstructures. We summarize device performances of spin-coated small-area devices of PM6:Y6 analogues in **Table S2** and GIWAXS characterization results of best cells in **Fig.S15**. We moved both supplementary tables to **Table R1** and **R2** and copied GIWAXS result to **Fig. R1** to facilitate your reading. From the results,

we could find the difference of V_{oc} between small and large area devices. Moreover, the difference of ΔV_{oc} , follows $\Delta V_{oc,N3} > \Delta V_{oc,L8-BO} > \Delta V_{oc,Y6}$. The ΔV_{oc} between the best small-area device and large-area device fabricated with PM6:Y6 is only 0.027 V. While for L8-BO, such value raises toward 0.048 V, nearly twice compared with the former. However, GIWAXS results unveil significant differences in packing styles in N3-based films, moderate changes in L8-BO-based films, and little changes in Y6-based films. This could once again prove the strong correlation between the packing style of acceptors and V_{oc} related loss in OPVs.

We have made following changes to the manuscript to address the concern:

At the end of first paragraph in “In-situ UV-vis Spectrum of Slot-die Film-forming Process”, starting on page 5: “When compared to small-area devices fabricated with spin-coating technique and o-xylene as a solvent, as shown in **supplementary table 2**, L8-BO achieves the best performance with both small and large area devices, Y6 obtain a mediocre result, and N3 excels in a small area but failed catastrophically in large area devices.”

At the end of the section “Multiscale structural analysis”, starting on page 15: “The influence of such packing attributes on device performance is significant. In **Figure S15** we analyzed and plotted GIWAXS results of the best spin-coated devices, which, qualitatively speaking, showed a heavy correlation between stacking change and performance deviance between small-area devices and large-area devices, where N3 systems showed the largest difference in packing styles and device performance, Y6 and L8-BO system changes less and thus retained most of their performance in small-area devices.”

At the end of “Assembly-controlled film forming kinetics” section, starting on page 23: “Following the analysis of our kinetics, it is now possible to discuss a bit more in detail about the performance deviation between small-area and large-area devices. Our kinetic model indicates an absorbing behavior of acceptor molecules onto the donor fibrils, which is generally bad for spacing between donor and acceptor and brings extra energy loss to the system. When focused on Y6 and L8-BO systems, we find out that loss in V_{oc} when scaling up for L8-BO systems is nearly doubled compared with Y6 systems (0.048 V vs 0.027 V). We attribute such extra loss to the packing style shift of L8-BO systems. From **Figure 3** and **Figure S15** we could tell that L8-BO tends to assemble in a more relaxed formation when being slot-die coated (smaller q vector peaks dominate compared with spin-coated films), which could induce more

coupling between charge transfer state and molecule vibronic states and thus brought excess energy loss.”

We’ve also added the fabrication conditions for small-area devices in the “method” section as follows:

“The rigid small-area devices were fabricated by spin coating with an inverted device structure of glass/ITO/ZnO/active layer/MoO_x/Ag. The patterned ITO glass was cleaned by sequential sonication in soap with deionized (DI) water, then in DI water, ethanol, and finally in isopropyl alcohol for 30 min each. After ultraviolet-ozone (Ultraviolet Ozone Cleaner, Jelight Company, USA) treatment for 15 min, the ZnO precursor was spin-coated on the ITO substrate at 4000 rpm to form an electron transporting layer ZnO. Then, the substrate was baked at 200 °C for 30 min. The Donor and acceptor materials were dissolved in an *o*-xylene solvent with a total concentration of 16.7 mg/ml (D:A=1:1.2 w/w). The active layers were spin-coated and post-processed in an N₂ glove box at room temperature following conditions listed in **supplementary table 2**. At a vacuum level of $\approx 1.0 \times 10^{-6}$ mbar, a thin layer (5 nm) of MoO_x was deposited as the anode interlayer. Finally, a 160 nm of Ag was deposited onto the active layer to form a back electrode. Photovoltaic performance of small-area devices were measured in a N₂-filled glovebox. Newport Thermal Oriel 91159 A solar simulator was used for *J-V* curves measurement under AM 1.5 G, and the light intensity was calibrated with Newport Oriel PN 91150 V Si-based solar cell. Typical cells have device areas of approximately 4 mm². A mask with well-defined area (2.56 mm²) was used in *J-V* characteristics as well. *J-V* measurement signals were recorded by Keithley 2400 source-measure unit.”

Fig.R1 2D GIWAXS image and in-plane peak splitting result of spin-coated PM6:Y6 analogue

films.

Table R1. Performance of 0.04 cm² OPV devices spin-coated with o-xylene.

Donor	Annealing	Rotation Speed (Rpm)	Acceptor	Voc (V)	Jsc (mA/cm ²)	FF (%)	PCE (%)	
PM6	110 °C 10min	3000	Y6	0.769±0.001 0.770	17.536±0.142 17.745	65.961±0.803 66.436	9.544±0.160 9.738	
			N3	0.800±0.001 0.800	20.171±0.151 20.021	68.980±1.826 67.154	11.934±0.398 11.536	
			L8-BO	0.845±0.007 0.850	21.487±0.391 21.863	74.260±0.406 74.605	14.464±0.453 14.870	
		3500	Y6	0.767±0.002 0.769	16.826±0.237 17.179	66.339±0.629 67.222	9.186±0.222 9.528	
			N3	0.802±0.002 0.803	20.035±0.304 20.275	74.481±0.373 74.412	12.832±0.169 12.997	
			L8-BO	0.848±0.003 0.847	22.412±0.392 22.675	75.508±0.474 75.964	15.396±0.269 15.647	
		4000	Y6	0.766±0.004 0.769	16.207±0.400 16.689	66.187±0.845 66.872	8.807±0.223 9.203	
			N3	0.800±0.003 0.799	19.337±0.500 19.453	71.655±1.588 73.044	11.889±0.498 12.181	
			L8-BO	0.849±0.003 0.851	22.228±0.272 22.619	75.204±1.353 76.938	15.230±0.455 15.878	
		w/o.	3000	Y6	0.772±0.005 0.777	16.777±0.174 16.887	64.546±0.350 65.133	8.962±0.182 9.171
				N3	0.826±0.002 0.827	19.227±0.210 19.294	71.869±1.296 73.062	12.240±0.325 12.506
				L8-BO	0.861±0.004 0.868	20.492±0.119 20.299	74.134±0.266 74.548	14.020±0.063 14.080
	3500		Y6	0.776±0.001 0.777	16.292±0.373 16.595	65.049±0.674 65.787	8.822±0.224 9.092	
			N3	0.821±0.002 0.820	19.367±0.215 19.698	71.072±1.759 71.625	12.119±0.347 12.414	
			L8-BO	0.860±0.002 0.862	21.382±0.216 21.606	74.333±0.485 73.995	14.659±0.154 14.773	
	4000		Y6	0.771±0.006 0.773	15.273±0.237 15.613	63.580±3.904 66.934	8.040±0.649 8.664	
			N3	0.817±0.004 0.821	17.294±0.632 18.120	70.848±1.548 72.641	10.747±0.598 11.589	
			L8-BO	0.866±0.002 0.869	22.110±0.036 22.126	75.394±0.205 75.653	15.490±0.080 15.600	

Table R2. Performance of 1 cm² OPV devices coated under various temperature.

Donor	Temperature (°C)	Acceptor	V_{oc} (V)	J_{sc} (mA/cm²)	FF (%)	PCE (%)
PM6	40	Y6	0.743±0.009 (0.756)	8.42±0.33 (8.48)	48.89±3.20 (52.29)	3.06±0.28 (3.35)
		N3	0.667±0.077 (0.662)	1.33±0.28 (1.95)	26.09±1.08 (26.67)	0.233±0.060 (0.344)
		L8-BO	0.810±0.009 (0.812)	20.65±1.02 (21.58)	59.71±4.14 (64.94)	10.00±0.98 (11.37)
	60	Y6	0.741±0.009 (0.747)	11.21±0.31 (11.5)	51.97±2.56 (54.5)	4.32±0.33 (4.68)
		N3	0.687±0.043 (0.738)	1.34±0.14 (1.62)	25.22±1.71 (28.94)	0.234±0.048 (0.346)
		L8-BO	0.804±0.008 (0.81)	20.46±0.61 (20.68)	64.56±5.24 (69.99)	10.61±0.88 (11.72)
	80	Y6	0.733±0.013 (0.755)	15.74±1.29 (17.61)	52.74±5.54 (60.63)	6.09±0.89 (8.05)
		N3	0.640±0.079 (0.776)	2.57±0.84 (4.2)	26.07±2.29 (29.3)	0.459±0.244 (0.954)
		L8-BO	0.799±0.004 (0.802)	20.68±0.84 (21.78)	66.94±1.71 (67.68)	11.06±0.52 (11.81)
	100	Y6	0.726±0.012 (0.743)	18.14±1.09 (19.46)	57.44±3.04 (60.06)	7.57±0.68 (8.69)
		N3	0.598±0.186 (0.755)	6.73±2.28 (9.82)	30.79±4.23 (32.56)	1.29±0.67 (2.41)
		L8-BO	0.807±0.004 (0.803)	21.28±0.84 (21.97)	67.18±1.80 (68.97)	11.53±0.36 (12.17)

2. A more detailed description of the in-situ measurement of the absorption spectra would be appreciated. Specifically, it would be beneficial if the authors could clarify whether the spectrum is measured in reflection mode or transmission mode. Furthermore, it is important for the authors to explain how the absorption spectrum is determined, whether it is calculated using the equation $A=1-R$, $A=1-T$, or $A=1-R-T$. Additionally, it would be helpful to specify whether the absorption spectrum is measured on devices or using organic blend films.

Response: Thank you for addressing your concern. Here are the answers to your questions:

1) *whether the spectrum is measured in reflection mode or transmission mode.*

Our in-situ UV-vis spectroscopy was performed using transmission mode, with one fiber spectrometer obtaining the transmission signal.

2) *how the absorption spectrum is determined, whether it is calculated using the equation $A=1-R$, $A=1-T$, or $A=1-R-T$.*

The absorption spectrum is calculated using equation $A=1-T$, where A is absorption and T is the recorded transmittance signal.

3) *it would be helpful to specify whether the absorption spectrum is measured on devices or using organic blend films.*

The absorption spectrums were collected on organic blend films following the procedures in the method section. We also collected an in-situ spectrum of the active layer in the fabrication process of devices with the same fabrication process as organic blends. A comparison between *in-situ* spectrum of organic blend films and the active layer of devices is shown in **Fig.R2**. Qualitatively speaking, we consider no major deviation between the film formation kinetics in types of films.

To clarify the concerns in our manuscript, we made the following changes:

On page 6, to the caption of Figure.1a: “**Instrumental setup of slot-die integrated *in-situ* UV-vis spectrometer which works on transmission mode**”.

On page 26, to the method section titled “Slot-die Process and In-situ UV-vis spectroscopy”: “**Transmission mode is used in the collection of in-situ spectroscopy signal.**” is added to the end of the first paragraph.

Fig.R2 Comparison of in-situ UV-vis spectroscopy between organic blend and active layer of a OPV device. Meaning of lines in follows that is shown in Fig.2 g-h.

3. *In this study, the authors summarize the structural formation process of active thin films containing acceptors with different side chains deposited via xylene. It would be beneficial for the authors to clarify whether the observed patterns of structural formation can also be applicable to blended films deposited using other solvents.*

Response: Thank you for addressing your concern on the generality of our findings in the manuscript. We would like to try more solvent systems and further validate our kinetic model,

but the experimental condition restricted our experiment with toxic halogen-containing solutions like chloroform and chlorobenzene, while PM6 seldom dissolves in tetrahydrofuran(THF). But still, we managed to analyze the film-forming process with UV-vis spectroscopy on films coated with toluene.

We summarized the findings put them in **supplementary note 5**. A brief discussion on results was added to the end of “Assembly-controlled film-forming kinetics” section as follows on page 23:

“Moreover, we performed the same UV-vis spectroscopy analysis on the same systems with toluene as solvent. As shown in **Figure S7** and **Figure S8**, it is astonishing to find that compared with o-xylene, toluene tends to facilitate aggregation of acceptors and thus leads to a higher portion of growing kinetics and crystallization. Details of the analysis can be found in **Supplementary Note 5**. The result is anticipated in our proposed kinetic model since toluene poses lower resistance packing with acceptor backbones compared with o-xylene, thus acting as a better lubricant and helping better in the assembly of acceptors.”

4. A higher coating temperature appears to improve device performance. I wonder if the authors could further increase the coating temperature to over 100°C to investigate whether significantly altering the coating conditions can lead to a change in the growth mode of the blended active layer.

Response: Thank you for suggesting on extending our experiment conditions. However, due to the thermal instability of the PET substrate, we could not further increase the coating temperature beyond 100°C. However, we managed to further investigate film-forming kinetics on glass substrates with a higher coating temperature of 115°C. Results of which are added to **supplementary Figure 9**. We’ve also tried coating under 130°C but the rapid evaporating solution had overrun our experimental setup, and we got no *in-situ* spectroscopy data for the rapidly changing phase.

From what we have collected, the higher coating temperature generally increases the growing portion of kinetics in all three blends. Especially for PM6:N3 system, the even-higher temperature further inhibited the over-crystallization, remedying baseline shift and deduction of acceptor absorption.

Such a phenomenon is anticipated, as we have already discussed in supplementary note 4. All three blends coated here have already passed the critical point where solvent evaporation and molecule diffusion reach equilibrium. The faster solvent evaporation limits the diffusion of molecules and thus reduces the ability of chain formation, yielding a higher portion of grow kinetics in our model.

To have your concern settled, we have added a brief discussion at the end of our “Assembly-controlled film-forming kinetics” section as follows on page 22:

“We further tried coating blends on glass substrates with a substrate temperature of 115°C, the results of which are shown in **Figure S9**. After fitting with our grow-move two peak model the portion of growing peaks have all increased, which is in accordance with our kinetics, further strengthening the liability of our kinetic model.”

5. It appears that the coating temperature affects both J_{sc} and FF of the solar cells based on Y6 and L8-BO, but has minimal impact on the FF of the solar cell based on N3, despite significant changes in the morphology of the active layer. It would be helpful if the authors could provide an explanation for this observation.

Response: Thank you for raising the concern. To answer in brief: a similar microstructure is the source of the near-constant fill factor of N3 system, while such one-phase morphology, according to previous research, is the source of its poor performance.

The morphology of PM6:N3 film, on a larger scale which could trigger observable Mie scattering effect, is significantly different from each other. However, on a scale of tens of nanometers, which can be proved by similar GIWAXS patterns and AFM images, the morphology of PM6:N3 films is similar.

In our kinetic model, similar morphology could result from a film where large quantities of N3 molecules had assembled and from large aggregates, while PM6 and remaining saturated-absorbed N3 molecules form the actual film. In this state, we could actually consider the active layer to have only one phase, plus non-film forming N3 crystals. According to Ye et.al. (Nat. Mater. 2018, 17, 253-260), a film composed of one single phase usually comes with a low fill factor.

6. *It is recommended to provide JV curves and EQE spectra for all the solar cells investigated in this study. This would enable a better understanding of the role played by the acceptor side chain in determining the morphology and performance of slot-die coated solar cells.*

Response: Thank you for raising the concern. Per your request, we have put all JV curves and EQE spectrums of the best cells for each coating condition in **Figure S10**.

7. *In the first paragraph of the results section, the statement "Later, AFM and GIWAXS results indicate large morphology differences in tens of Angstroms and larger scales" is unclear and requires rephrasing. It is unclear what is meant by "morphology differences in tens of Angstroms."*

Response: Thank you for raising the concern on rephrasing the expression of our manuscript. We have modified the confusing sentence (on page 4) into:

“Later, GIWAXS and AFM results suggest that the morphology of coated PM6/Y6 analogue films have significant differences on a scale from several molecules to aggregates with a few hundred nanometers.”

8. *In the second paragraph of the results section, it is mentioned that “PCEs of 1 cm² devices made from PM6/Y6 analogues are dramatically different in numbers and temperature dependency”. The temperature dependency mentioned here needs to be further clarified. It should be specified that this temperature refers to the substrate temperature rather than the temperature of the slot-die head or any post-annealing temperature.*

Response: Thank you for finding this blurring expression in our manuscript. To address the problem, we have modified our manuscript on page 5 as follows:

“PCEs of 1 cm² devices made from PM6/Y6 analogues are dramatically different in numbers and dependency on the substrate temperature, as shown in the table and **Figure S10”**

9. *In the first paragraph of the discussion section, it is stated that “our capability in the manipulation of nanoscale morphology much stronger than before, such as targeted induction or inhibition of molecular assembly either by intentional molecular design or selective addition of certain additives.” The authors should provide further clarification on how the results of this*

study contribute to the improvement of the selection rule for additives.

Response: Thank you for requesting detailed discussion on the potential of our kinetic model and we appreciate your high value on our manuscript. To further clarify the selection rule of additives, we add the following discussion to the end of the first paragraph of the discussion section on page 24:

“By understanding the primary assembly of acceptor molecules, additive molecules which target certain binding sites can be designed and synthesized to bind with acceptors, encouraging or discouraging certain types of assembly, and guiding the assembly toward another direction. For example, suppose a conjugated solid additive can be synthesized to target the core acceptor unit of Y6 and N3 molecule; such binding will reduce the formation of the star-shaped trimer, thus forcing the kinetics to move toward moving kinetic. Moreover, the host-guest strategy might also be applied to facilitate more favorable assembly patterns and multiscale morphology in the formation of active layers.”

Reviewer #3 (Remarks to the Author):

In this manuscript, Zhang et al. did a fascinating work by incorporating multiple experimental procedures and simulations to unveil the critical assembly structure and film-forming kinetics of famous Y6 analogues in slot-die-coated large-area organic photovoltaics. Each of the used characterization techniques was broadly applied to the topic of organic photovoltaics, but the authors made an inspiring elucidation on how to combine them to yield more profound insights. Among these, the proposition of a star-shaped trimer is the most fascinating part of the work, where authors did their best to concretize the assembly structure, via both in-situ spectroscopy and GIWAXS characterization. The evidence presented in this manuscript is convincing enough to support the proposed structure and the following kinetic model. I recommend the manuscript be published in Nature Communications after minor revisions. Several concerns that could be

addressed are listed as follows:

Response to comment: We are grateful to the reviewer for the positive comments to our work and high thoughts on the manuscript. Especially the agreement on our experimental procedures developed in the manuscript.

1.The presented UV-vis spectrums in Fig. 2 have significant differences compared with each other. But since the authors only showed one in-situ UV-vis spectrum for each condition (including that in supporting information), this raises the concern of repeatability of the observed phenomenon. It would be much better if the author could provide extra information on the repeatability of UV-Vis spectroscopy.

Response: Thank you for raising your concern. We have to clarify that each experimental condition has been reproduced at least three times to avoid mistakes. Moreover, the statistical analysis in Fig.S17 is exactly the result of this repeated work. In the revised version of our manuscript, please find enclosed spreadsheets for all raw fitting results of our model used to draw **Figure S18** in document S2 of supplementary information. To further prove the repeatability of our experiment, we put the result of the repeating experiment of **Fig.2a-c** in **Fig.R3** for your reference. Though the actual time between stage I and stage II varies from experiment to experiment, which is brought by the manual start of spectrum acquisition, the shape in the waterfall plot and the fitted result showed good repeatability. We believe such repeatability could satisfy the requirement of accuracy in our current manuscript.

Fig.R3 Result of repeated experiment of PM6:Y6 analogues with substrate temperature set at 40°C.

2.The authors mentioned the formation of nanocrystals and their size dependence on coating temperature. But no other experimental evidence except for the baseline shift in UV-vis spectroscopy is provided. Further evidence is highly desirable if the authors would like to make this claim.

Response: Thank you for raising your concern. To address the problem, we've supplemented our claim by optical microscope images of PM6:N3 blend film in **Figure S11**. We've also put the images here for your reference as **Fig.R4**. Moreover, we've modified our expression in the manuscript discussing baseline shift on page 7 as:

“... and thus baseline declined again. As depicted in **Figure S11**, the optical microscope of PM6:N3 blend film exhibited clear temperature-dependent crystallization, which could further be the evidence of our discussion above.”

Fig.R4 Optical microscope image of PM6:N3 blend film coated under various substrate temperature.

3.The AFM image of PM6 in Fig.3a is unprecedented in previous reports. Authors should perform the AFM characterization again to further investigate the origin of such a distinguished branch-like structure. Also, authors should add a color bar to the AFM images.

Response: Thank you for raising your concern. We have prepared pure PM6 films for new AFM imaging. It turned out that the branch-like structure could be the batch variance for PM6 since

we could also observe a similar structure in the AFM image of blend films. But since a similar structure could not be reproduced, we would not conduct further analysis on such structure and replaced the original AFM image of PM6 film with new results. The discussion on AFM images in our manuscript on page 11 have also been modified as following:

“The morphology of pure PM6 film and three blend films was in great disagreement at first glance. Pure PM6 film exhibited a clear fibril feature, while in the Y6 blend’s film, a spherical structure is observed with greater fluctuation in height. N3’s blend showed a similar fibril structure compared with that of PM6 film while the L8-BO blend exhibited a flake-like structure.”

Also, thank you for pointing out the missing color bar problem when we process our AFM image, we thus redraw all our AFM images and added a color bar to each of them.

4. In Supplementary Note 3 and Fig.S11, authors brilliantly designed feature vectors for automated packing analysis of molecular dynamics trajectory. Such a tool is critical to the discovery of star-shaped trimers and other assembly forms from a large quantity of molecular dynamic trajectories. However, the definition and the calculation method of the principal axis seem to be missing from the supplementary note. The authors should further elucidate the procedure to calculate the principal axis, as well as the generation of mentioned feature vectors.

Response: Thank you for raising your concern. To further elucidate, we will answer your question in the following two topics:

- Calculation procedure of principal axis for an acceptor molecule:
 - 1) Determine the geometrical center for terminal groups of an acceptor molecule. In our case, terminal group refers to 2-(5,6-difluoro-2-methylene-3-oxo-2,3-dihydro-1H-inden-1-ylidene)malononitrile units of Y6 analogues.
 - 2) Set the line determined by the two geometrical centers as X-axis of the principal axis, and the midpoint of the line as principal center.
 - 3) Determine Y axis. The Y axis would be the line in such a plane that: I. contains both geometrical centers, II. the summation of distance those backbone atoms to the plane reaches the minimum. Then, the Y axis will be the perpendicular line that intersect X axis at principal center.
 - 4) Calculate Z axis. The Z axis is finally determined by:

$$\vec{z} = \vec{x} \times \vec{y}$$

We've also added the guidance above to supplementary note 3 for better guidance to the readers of our manuscript.

- Generation of feature vectors:

As for the generation of feature vectors, we would like to publicize our python code used to process molecule coordinate data and generate feature vectors for your detailed reference.

The publicized code is altogether stored in the same repository of the dataset, named `calc_feature_vec.py`.

5. The analytical method used in the manuscript could be inspiring to the community. Moreover, the ML model used to automatically classify the packing motif in a molecular dynamics trajectory would be valuable in many more scenarios. We here suggest authors publicize the model and the generated dataset for further applications.

Response: Thank you for valuing our model and dataset. It is our honor to take your advice sharing our model and dataset. We had created a public repository on GitHub which keeps the dataset, code to process the feature vector, and one simple example on how to use the dataset. Also, we had added and modified our data availability section and code availability section as follows:

“Data Availability

Datasets generated and used for the analysis of molecular dynamic trajectories are stored in the GitHub repository:

<https://github.com/PulsEternal/PackingAnalysisY6analogues>.

All fitting results of conducted in-situ UV-vis spectroscopy experiments, which are also the raw data used to generate the statistical plot of **Figure S18** can be found in Document S2.

Other data that support the findings of this study are available from the corresponding author upon reasonable request.

Code Availability

Code used to generate feature vectors used in the analysis of molecular dynamic trajectories are stored at the GitHub repository:

<https://github.com/PulsEternal/PackingAnalysisY6analogues>”

REVIEWERS' COMMENTS

Reviewer #1 (Remarks to the Author):

The authors addressed all my concerns and the manuscript can be accepted.

Reviewer #2 (Remarks to the Author):

The authors have carefully addressed all of my comments. I do not have any further comments on the manuscript.